# Regulation of Transporters for Organic Cations by High Glucose

**DOI:** 10.3390/ijms241814051

**Published:** 2023-09-13

**Authors:** Martin Steinbüchel, Johannes Menne, Rita Schröter, Ute Neugebauer, Eberhard Schlatter, Giuliano Ciarimboli

**Affiliations:** Experimental Nephrology, Department of Internal Medicine D, University Hospital Münster, 48149 Münster, Germany; martin.steinbuechel@uni-muenster.de (M.S.); johannesmenne@web.de (J.M.); ritas@uni-muenster.de (R.S.); ute.neugebauer@ukmuenster.de (U.N.); schlate@uni-muenster.de (E.S.)

**Keywords:** organic cation transporters, glucose, regulation, diabetes

## Abstract

Endogenous positively charged organic substances, including neurotransmitters and cationic uremic toxins, as well as exogenous organic cations such as the anti-diabetic medication metformin, serve as substrates for organic cation transporters (OCTs) and multidrug and toxin extrusion proteins (MATEs). These proteins facilitate their transport across cell membranes. Vectorial transport through the OCT/MATE axis mediates the hepatic and renal excretion of organic cations, regulating their systemic and local concentrations. Organic cation transporters are part of the remote sensing and signaling system, whose activity can be regulated to cope with changes in the composition of extra- and intracellular fluids. Glucose, as a source of energy, can also function as a crucial signaling molecule, regulating gene expression in various organs and tissues. Its concentration in the blood may fluctuate in specific physiological and pathophysiological conditions. In this work, the regulation of the activity of organic cation transporters was measured by incubating human embryonic kidney cells stably expressing human OCT1 (hOCT1), hOCT2, or hMATE1 with high glucose concentrations (16.7 mM). Incubation with this high glucose concentration for 48 h significantly stimulated the activity of hOCT1, hOCT2, and hMATE1 by increasing their maximal velocity (V_max_), but without significantly changing their affinity for the substrates. These effects were independent of changes in osmolarity, as the addition of equimolar concentrations of mannitol did not alter transporter activity. The stimulation of transporter activity was associated with a significant increase in transporter mRNA expression. Inhibition of the mechanistic target of rapamycin (mTOR) kinase with Torin-1 suppressed the transporter stimulation induced by incubation with 16.7 mM glucose. Focusing on hOCT2, it was shown that incubation with 16.7 mM glucose increased hOCT2 protein expression in the plasma membrane. Interestingly, an apparent trend towards higher hOCT2 mRNA expression was observed in kidneys from diabetic patients, a pathology characterized by high serum glucose levels. Due to the small number of samples from diabetic patients (three), this observation must be interpreted with caution. In conclusion, incubation for 48 h with a high glucose concentration of 16.7 mM stimulated the activity and expression of organic cation transporters compared to those measured in the presence of 5.6 mM glucose. This stimulation by a diabetic environment could increase cellular uptake of the anti-diabetic drug metformin and increase renal tubular secretion of organic cations in an early stage of diabetes.

## 1. Introduction

As per the theory of remote sensing and signaling, the control of systemic and localized concentrations of endogenous low-molecular-weight compounds entails a finely orchestrated interplay among transporters, metabolizing enzymes, and regulatory proteins distributed throughout diverse organs. This intricate interplay gives rise to a complex communication network that operates not only within organs but also extends to interactions between different organisms [1]. Within the framework of the remote sensing and signaling system, certain entities possess cationic attributes, thereby rendering them amenable to transportation by organic cation transporters. This class of substances encompasses polyamines and their metabolites, including but not limited to cadaverine, putrescine, spermine, spermidine, acrolein, and trimethylamine oxide, as well as guanidino compounds such as creatinine, guanidine, and methylguanidine [2]. These cationic substances are mainly eliminated from the body through renal secretion, which is mediated by highly expressed organic cation transporters (OCTs) and multidrug and toxin extrusion proteins (MATEs) in the kidneys (Figure 1). In cases of renal failure, there is a significant increase in the circulating concentration of these organic cations, leading to potential damage to other organs, and for this reason these substances are called uremic toxins. Uremic toxins, in turn, can adversely affect the functioning of other organs such as the heart [3], brain [4], and the liver [5]. In the liver, OCTs and MATEs jointly establish the directional transport mechanism responsible for the hepatic elimination of organic cations (Figure 1).

The directional transport of organic cations through the OCT/MATE system holds significant pharmacological significance, considering that roughly 40% of prescribed drugs fall under the category of organic cations [7]. These drugs are eliminated from the body through hepatic and/or renal secretory clearance mechanisms facilitated by the OCT/MATE system [8]. As specific examples, the antidiabetic medication metformin [9,10] and the muscarinic antagonist trospium [11,12] are both substrates of this clearance system. Interestingly, the activity of the OCT/MATE system is well recognized to be subject to both rapid and long-term regulation. Indeed, various factors can influence the expression and function of these transporters, leading to alterations in their activity levels [13,14,15]. Therefore, the regulation of the OCT/MATE system can have significant physiological and pharmacological implications. Glucose, in particular, emerges as a molecule of particular interest due to its multifaceted role in the body. As the primary source of energy for all organisms, glucose plays a crucial role in cellular metabolism. Its concentration is tightly controlled to maintain stable blood glucose levels and ensure proper energy supply to tissues and organs. Disruptions in glucose homeostasis can lead to various metabolic disorders, including diabetes with its complications. Furthermore, glucose can serve as an essential signaling molecule that can modulate the expression of numerous genes in various tissues and organs of mammals [16]. This regulatory phenomenon is particularly intriguing, as it underscores glucose’s potential significance as a pivotal molecule in modulating the activity of transporters associated with the remote sensing and signaling systems. Glucose can act as an integrative factor, orchestrating the function of various members within these systems, including OCTs and MATEs, distributed across diverse organs. Hyperglycemia is a typical feature of diabetes mellitus and is probably involved in regulation of transporters. Indeed, uncontrolled diabetes mellitus changes the expression of renal transport proteins in rats [17,18]; more specifically, OCT expression appears to be downregulated in experimental diabetes [19,20,21]. To our knowledge, there is still no information on MATE expression under diabetic conditions.

In this work, we investigated whether the expression and function of human OCT1 (hOCT1), hOCT2, and hMATE1 are affected by high glucose levels in human embryonic kidney 293 (HEK293) cells stably expressing these transporters. The activity of the transporters was assessed by measuring the uptake of the fluorescent organic cation 4-(4-(dimethylamino)styryl)-N-methylpyridinium (ASP^+^), as described in the Section 4.

## 2. Results

Incubation of hOCT2-HEK293 cells for 24 h with increasing glucose concentrations (5.6–16.7 mM) did not alter the initial ASP^+^ uptake (see Figure 2).

Therefore, the incubation time with glucose was extended to 48 h. Under this condition, a significant increase in hOCT2 activity was measured for all glucose concentrations compared to control experiments performed in the presence of 5.6 mM glucose (see Figure 3a). For hMATE1, a significant increase in activity was only observed after 48 h of incubation with 16.7 mM glucose (see Figure 3b). Therefore, the following experiments were performed after 48 h incubation with 5.6 or 16.7 mM glucose concentrations of HEK293 cells stably expressing hOCT1, hOCT2, or hMATE1, followed by measurement of ASP^+^ uptake.

To identify the regulatory mechanism of glucose on ASP^+^ uptake, saturation experiments with 0–20 µM ASP^+^ concentrations were performed in hOCT1-, hOCT2-, or hMATE1-HEK293 cells after 48 h incubation with 5.6 and 16.7 mM glucose. “Non specific” ASP^+^ uptake was assessed in the presence of a large excess (1 mM) of tetrapentylammonium (TPA^+^), an effective inhibitor of hOCT1 [22], hOCT2 [2,23], and hMATE1 [13]. The specific transporter-mediated ASP^+^ uptake was calculated by subtracting the “non specific” from the total uptake. Figure 4 shows an example of these experiments using hOCT2-HEK293 cells. This approach allows the determination of ASP^+^ uptake kinetic parameters (K_m_ and V_max_) after 48 h incubation with 5.6 and 16.7 mM glucose. The regulation by 16.7 mM glucose did not depend on an increased osmolarity of the incubation solution, since in experiments in which the 5.6 mM glucose incubation solution was supplemented with mannitol to a total hexose concentration of 16.7 mM, no change in ASP^+^ uptake rates was detected after 48 h incubation (Appendix A). Mannitol is considered to be an impermeable hexose compared to glucose.

One of the most important cellular glucose-sensing pathways is the mechanistic target of rapamycin (mTOR) kinase, which forms the mTORC1 and mTORC2 protein complexes [24]. Under physiological glucose concentrations, mTORC1 is activated. Conversely, when glucose levels are low, mTORC1 is inhibited. In diabetes, mTORC1 activity is increased in hepatocytes [25] and in renal proximal tubule cells [26]. In these cells, high levels of hOCT1 [27], hOCT2, and hMATE1 are specifically expressed [27,28].

Torin-1 is a small molecule, which binds to the adenosine triphosphate (ATP) binding site of mTOR and significantly inhibits mTORC1 and mTORC2 activity [29]. Therefore, we investigated whether the regulation of organic cation transporters observed under high glucose incubation is dependent on mTOR activity by performing the ASP^+^ uptake experiments in the presence of Torin-1. First of all, the effect of 48 h incubation with 16.7 mM glucose on hOCT1-mediated ASP^+^ uptake was investigated in the presence of 10, 100, and 1000 nM Torin-1 (Figure 5). Since a 100 nM Torin-1 concentration was the lowest concentration that significantly affected ASP^+^ uptake, this concentration was used for all the experiments. Torin-1 did not alter the non-specific ASP^+^ uptake determined in the presence of a large excess of TPA^+^ (1 mM, Appendix A of Appendix A).

Figure 6 summarizes the effects of 48 h incubation with 16.7 mM glucose on specific ASP^+^ uptake measured in the presence or absence of 100 nM Torin-1 using hOCT1- (panel a), hOCT2- (panel b), and hMATE1- (panel c) HEK293 cells. Incubation with 16.7 mM glucose significantly altered the V_max_ in each cell line without affecting the K_m_ value of ASP^+^ uptake (Table 1). Nevertheless, it is important to highlight that the K_m_ for ASP^+^ uptake in hOCT1-HEK293 cells exhibits a twofold increase after 48 h of incubation with 16.7 mM glucose compared to what was observed under 5.6 mM glucose conditions, even though this difference did not reach statistical significance. Incubation with 100 nM Torin-1 completely suppressed these effects.

To elucidate the mechanisms underlying the transporter regulation by high glucose incubation, the mRNA expression of the transporters for organic cations under investigation was compared after 48 h incubation with 5.6 or 16.7 mM glucose (Figure 7).

Since the larger increase in mRNA expression by high glucose was detected in hOCT2-HEK293 cells (Figure 7), the expression of hOCT2 in the plasma membrane was investigated by biotinylation experiments of hOCT2-HEK293 cells after 48 h incubation with 5.6 or 16.7 mM glucose. As shown in Figure 8, a significant increase in hOCT2 membrane expression was detected after 48 h incubation with 16.7 mM glucose compared to experiments performed with 5.6 mM glucose. Western blot analysis of the biotinylation experiments is shown in Appendix A.

In further experiments, the effect of 48 h incubation with 5.6 or 16.7 mM glucose was investigated in Madin–Darby Canine Kidney (MDCK) cells expressing hOCT2 or the empty vector (EV) (Figure 9). These cells were grown in an extracellular matrix to form three-dimensional (3D) cysts, as previously described [30]. In this setup, hOCT2 is clearly expressed in the basolateral membrane domain of the cysts and is easily accessible for experiments. Since HEK293 cells are not polarized, the transport of organic cations in this cell model is not vectorial, unlike the physiological situation in hepatocytes or proximal tubular cells. Using this MDCK cysts system, it was possible to investigate whether the glucose regulation observed in the experiments described so far in the non-polarized HEK293 cells is also present using a polarized cell expression system, in which transporters are expressed in the basolateral membrane domain, similar to their physiological expression in vivo [31,32]. In MDCK cells expressing the empty vector, a tiny uptake of ASP^+^ was detected, which was not different after 48 h incubation with 5.6 or 16.7 mM glucose. Conversely, in MDCK cells expressing hOCT2, a stronger ASP^+^ uptake was observed, which was significantly increased after 48 h incubation with 16.7 mM glucose.

The mRNA expression of hOCT2 was measured by quantitative PCR analysis in 13 kidney samples from patients who had one kidney removed because of cancer. Three of these samples were from patients with diabetes mellitus. Comparison of quantitative hOCT2 mRNA expression shows a trend towards higher hOCT2 expression in samples from diabetic patients (Figure 10). However, due to the small number of samples analyzed, these results must be interpreted with caution. Unfortunately, we were unable to obtain more samples from diabetic patients.

## 3. Discussion

Transporters for organic cations play important physiological and pharmacological roles. They are part of the remote sensing and signaling system [33] and mediate the excretion of potential endogenous toxins [2] and of many drugs [34]. The activity of transporters for organic cations is subject to rapid regulation, which can be mediated by several different signaling pathways [13,30,35,36] and chronic regulation that can be induced by various pathological conditions such as diabetes [21,37], fibrotic diseases [38], and interventions such as organ transplantation [14]. Glucose is an important source of energy [39], but also has important roles as a regulator of various intracellular signaling pathways [16,40,41,42] and transporter functions [43]. Therefore, in this work we investigated the regulation of transporters for organic cations by glucose. A glucose concentration of 5.6 mM was chosen as “normal” because it is in the middle of the diabetes therapeutic target range and it is within the normal fasting non-diabetic range [44,45]. The activity of hOCT2 was significantly stimulated by 48 h incubation in the presence of glucose concentrations ranging from 8.9 to 16.7 mM. The hMATE1 showed a lower sensitivity to glucose stimulation, reaching a significant stimulation only by incubation with 16.7 mM glucose. Therefore, in all the other experiments, the effect of 16.7 mM glucose was compared with that measured after incubation with 5.6 mM glucose. Incubation with 16.7 mM glucose significantly increased the V_max_ of hOCT1, hOCT2, and hMATE1, without changing their affinity for transporting the fluorescent organic cation ASP^+^. These effects are mediated by mTOR activation, as they are suppressed by Torin-1, an mTOR inhibitor. The stimulation of transporter activity is probably caused by increased transcriptional activity, as shown by the significantly higher transporter-mRNA expression after incubation with 16.7 mM glucose. Activation of mTOR by nutrients is known to stimulate cellular transcriptional activity and protein expression [46]. Indeed, biotinylation experiments performed on hOCT2-HEK293 cells showed that incubation with 16.7 mM glucose increased hOCT2 expression in the plasma membrane. Conversely, previous studies investigating the effect of high glucose on the transport of organic cations showed different results. Incubation of CaCo_2_ cells (a human cell line used to model enterocytes) with high glucose (25 mM) reduced the affinity of hOCT3 for its substrates and its mRNA expression [47]. It should be noted that in the cited study, the cells were adapted to the high glucose concentration for at least five passages, a much longer time than that used in our study. In another study on rat mandibular osteoblasts, incubation of the cells with 16.5 mM glucose for 7 days increased the uptake of organic cations, similarly to what was observed in the present study. However, incubation with 16.5 mM glucose induced a phosphorylation of OCT1 with increased affinity for the substrates compared to what was measured with 5.5 mM glucose [48], in contrast to what was measured in the present study, where transporter activity stimulation was linked to changes of transporter V_max_.

To investigate whether the effects observed here were dependent on the expression system, the effects of 48 h incubation with 5.6 or 16.7 mM glucose were compared in MDCK expressing hOCT2 or the empty vector. Incubation with 16.7 mM glucose for 48 h increased the activity of hOCT2 expressed in MDCK cells grown in Matrigel. Under these conditions, MDCK cells form cysts, with hOCT2 expressed in the basolateral membrane domain, which is easily accessible to experimental solutions allowing a vectorial transcellular transport [30]. These results suggest that glucose stimulation of organic cation transport is also measurable in polarized cells with distinct apical and basolateral membrane domains, similar to the physiological situation in renal proximal tubule cells.

Hyperglycemia is a typical feature of diabetes mellitus. Therefore, hOCT2 mRNA expression in human kidneys was compared in apparently normal kidney tissue obtained from non-diabetic and diabetic cancer patients. These tissues were obtained after tumor nephrectomy and were taken from apparently healthy regions of the kidney. Unfortunately, we only had three samples from diabetic patients, and this number cannot be increased due to new surgical procedures that remove only tumor tissue, sparing apparently healthy adjacent tissue. Nevertheless, hOCT2 mRNA expression in kidney tissue from diabetic patients tends to be higher than that measured in non-diabetic patients, suggesting a translational relevance of the results obtained in cell culture. The very small number of patients with diabetes makes statistical analysis questionable. Certainly, our findings in the cell model cannot be directly extrapolated to the in vivo situation. The experiments with the HEK293 and MDCK cell lines only investigated the effect of high glucose over a relatively short period of a few days. In the in vivo situation, however, hyperglycemia or even diabetes mellitus is a long-term condition that lasts for weeks or months. Moreover, diabetes mellitus is a complex endocrine disease that involves not only high blood glucose levels, but also has many other metabolic consequences. For example, glucose is involved in the formation of advanced glycation end products (AGEs) and oxidative stress [49,50]. Increased oxidative stress also leads to increased generation of AGEs. The binding of AGEs to the RAGE (Receptor for AGEs) leads to the activation of numerous enzymes such as NADPH oxidase, which in turn increases the formation of new reactive oxygen species [49]. Thus, a vicious circle is formed in which the transcription factor Nf-kB is ultimately activated [49,50]. These conditions can completely change OCT regulation. In fact, animal models of diabetes mellitus have shown a reduction in OCT expression and function in diabetic animals [19,20,21,51].

Interestingly, the early stages of diabetes mellitus are characterized by hyperfiltration in the glomeruli and hypertrophy of the proximal tubules [52]. This is followed by a decline in renal function and kidney damage [52]. Hence, it is conceivable that, apart from glomerular hyperfiltration, hyperglycemia during the initial stages of diabetes mellitus could potentially enhance proximal tubule secretion through the hOCT2/hMATE1 axis, consequently augmenting the overall renal ion excretion. Interestingly, incubation with high glucose concentrations also stimulated organic cation transport and OCT2 expression in INS-1 cells, a model for pancreatic ß cells [53]. Stimulation of OCT activity by high glucose may also favor the anti-diabetic effects of drugs such as metformin, which is an OCT substrate, by increasing its uptake into target cells such as hepatocytes and pancreatic β-cells, at least in the early stages of the disease. Chronic kidney disease (CKD), a common comorbidity of diabetes, can lead to reduced renal function and hinder the excretion of metformin. Consequently, there may be a risk of metformin accumulating in the liver due to the possible glucose-induced upregulation of OCT1. This accumulation raises concerns regarding the potential development of lactic acidosis, a serious medical condition associated with metformin use in patients with CKD [54]. In conclusion, glucose may be a molecule that regulates the expression and function of organic cation transporters in several organs, and thus may be an important substance involved in the remote sensing and signaling system.

## 4. Materials and Methods

### 4.1. Cell Culture

For the experiments, human embryonic kidney (HEK) 293 cells stably expressing hOCT1, hOCT2, or hMATE1 were utilized. The generation of these cell lines has been previously described in other publications [13,55,56]. The HEK293 cells were maintained in 50 mL cell culture flasks (Greiner, Frickenhausen, Germany) at 37 °C with 5% CO_2_. The cell medium consisted of Dulbecco’s minimal Eagle’s medium (Biochrom, Berlin, Germany), supplemented with 10% fetal bovine serum, 1 g/L glucose, 2 mM glutamine, 3.7 g/L NaHCO_3_, and 100 U/mL streptomycin/penicillin (Biochrom). To ensure the selection of cells transfected with hOCT1, hOCT2, or hMATE1 transporters, geneticin (0.8 mg/mL, hOCT1/2-HEK293 cells) or hygromycin (200 mg/mL hMATE1-HEK293 cells) was added accordingly. The cells were cultured in 96-, 24-, or 12-well plates until they reached 80–90% confluence before conducting the experiments. The experiments were carried out using cells from passages 40–65. For specific experiments, Madin–Darby Canine Kidney (MDCK) cells stably expressing hOCT2 or the empty vector (EV) were grown in an extracellular matrix to form three-dimensional (3D) cysts, as previously described [30]. This type of cell culture, known as cyst-forming 3D culture, allows direct access to hOCT2 as it serves as the primary transporter for organic cations in the basolateral membrane domain. Detailed information about this setup is available in the publication by Koepp et al. [30]. The culture and functional analyses of these cells were approved by the state government’s Landesumweltamt Nordrhein-Westfalen of Essen, Germany (no. 521.-M-1.14/00).

### 4.2. Measurement of Transporter Function

In these experiments, the fluorescent organic cation ASP^+^ was used as a substrate to assess the function of hOCT1, hOCT2, and hMATE1 transporters [23,57]. ASP^+^ exhibits a shift in its emission spectrum from 515 to 580–590 nm upon entry into cells when excited at 450 nm, allowing for dynamic measurement of its cellular accumulation with high time resolution using a microfluorescence plate reader (Infinite F200, Tecan, Männedorf, Switzerland) [23]. The initial slope of cellular ASP^+^ uptake was measured to assess the transporter’s activity, excluding any potential impact from subsequent subcellular ASP^+^ distribution, efflux, or photobleaching [23].

To evaluate transporter function, cellular fluorescence was measured before and after the addition of ASP^+^ in the HEK cells expressing the transporters or hOCT2-MDCK and EV-MDCK cysts. The measurements were performed at a temperature of 37 °C. Specific transporter-mediated ASP^+^ uptake was assessed by subtracting the uptake measured in the presence of a high (1 mM) concentration of tetrapentylammonium (TPA^+^), a high-affinity inhibitor of transporter-mediated cellular ASP^+^ accumulation, from the total uptake. The regulation of ASP^+^ uptake by 24 and 48 h of incubation with increasing glucose concentration was investigated in hOCT2- or hMATE1-expressing HEK cells using 1 and 10 µM ASP^+^, respectively. A significant stimulation of ASP^+^ uptake was observed after 48 h of incubation with 16.7 mM glucose. To determine the affinity of the transporters for ASP^+^, saturation experiments were conducted, where specific ASP^+^ uptakes were measured in the presence of increasing concentrations of ASP^+^ (ranging from 0 to 20 µM) after 48 h incubation with either 5.6 or 16.7 mM glucose. The effects of 48 h of incubation with 16.7 mM glucose were also investigated in MDCK cysts, using 4 µM ASP^+^ as a transport marker. Additionally, the dependence of ASP^+^ uptake regulation by osmolarity was examined by adding an equimolar concentration of mannitol instead of glucose (see Appendix A). Mannitol is considered to be an impermeant hexose compared to glucose [42]. Appendix A shows that Torin-1 did not change unspecific ASP^+^ uptake. Appendix A shows the effects of Torin-1 on ASP^+^ uptake after incubation with 5.6 or 16.7 mM glucose in hOCT1-, hOCT2-, and hMATE1-HEK293 cells. All substances and standard chemicals were obtained from Sigma-Aldrich (Taufkirchen, Germany), unless indicated otherwise.

### 4.3. Real-Time PCR Analysis

In this study, total RNA was extracted from hOCT1-, hOCT2-, and hMATE1-expressing HEK293 cells after 48 h of incubation with either 5.6 or 16.7 mM glucose using the RNeasy Mini Kit (Qiagen, Hilden, Germany). The isolated RNA was further purified using a RNeasy column (Qiagen). For cDNA synthesis, 2 µg of the purified total RNA was utilized with the SuperScript III First-Strand Synthesis SuperMix (Invitrogen, Karlsruhe, Germany).

To analyze the gene expression profiles of hOCT1, hOCT2, and hMATE1, real-time PCR was performed using SYBR Select Master Mix for CFX (Thermo Fisher, Waltham, MA, USA) on a CFX Realtime Detection System (Biorad, Hercules, CA, USA). Specific primer pairs, as listed in Appendix A of the Appendix A, were used for amplification. Relative gene expression levels were determined using the 2^−ΔCt^ method, as described by Livak and Schmittgen [58]. GAPDH was used as the housekeeping gene for normalization. This method allows for the quantification of the gene expression levels relative to a reference gene (GAPDH) and the different glucose concentrations (5.6 and 16.7 mM) used in the incubation. Appendix A of Appendix A shows the Ct values for transporters and GAPDH in the cell lines used in this study.

### 4.4. Biotinylation of Cell Surface Proteins

Biotinylation of cell surface proteins was used to isolate plasma membrane proteins from hOCT2-HEK293 cells after 48 h incubation with 5.6 or 16.7 mM glucose. The Pierce Cell Surface Protein Isolation Kit (Thermo Scientific, Rockfort, IL, USA) was used according to the instructions of the manufacturer. Cells were washed two times with phosphate buffered saline (PBS) at 4 °C and incubated for 30 min with biotin at 4 °C. A quenching solution was used to terminate the reaction, and cells were washed and lysed with lysis buffer. After addition of a protease inhibitor (Roche Applied Science, Mannheim, Germany), cells were centrifuged and then incubated for 60 min on a NeutrAvidin column (Thermo Scientific). The columns were washed, and cell surface proteins were collected using sodium dodecyl sulfate-polyacrylamide gel electrophoresis (SDS-PAGE) buffer containing dithiothreitol (DTT). Samples for the measurement of total proteins were separated before incubation on the NetrAvidin column and resuspended in SDS-PAGE buffer. The samples were transferred to an SDS-PAGE gel (Mini-PROTEAN TGX GEL, Biorad) together with electrophoresis buffer. Electrophoresis was performed for one hour at 100–160 V. The gel was then blotted for 1 h at 100 V on a polyvinylidene difluoride (PVDF) membrane (Roche Applied Science). The PVDF membrane was incubated for 5 min in 3% gelatin to block unspecific binding and incubated overnight with mouse anti hOCT2 antibody (kindly provided by Prof. Koepsell) at a 1:250 dilution. The PDVF membrane was incubated for 45 min with horseradish peroxidase (HRP) coupled with goat-anti-mouse antibody (Dako, Hamburg, Germany) at a 1:5000 dilution and washed again. Immunoreactive bands were detected by enhanced chemiluminescence. The Western blot analysis of biotinylated samples is shown in Appendix A of Appendix A.

### 4.5. Human Samples

Human kidney samples were collected from both male and female Caucasian patients who underwent tumor nephrectomy at the Department of Urology, University Hospital Münster, Germany. The study protocol was approved by the Ethics Committee of University Hospital Münster, Germany, under the ethics vote 2020-858-f-S on 19 February 2021. Immediately after nephrectomy, a portion of normal kidney tissue, located away from the tumor and showing no signs of necrosis, was surgically removed. The collected tissue samples were placed in chilled phosphate buffer without bicarbonate (HCO_3_^−^ free) to preserve their integrity. Prior to sample collection, written informed consent was obtained from all the patients, ensuring that they were fully aware of the study’s purpose and procedures. To maintain the integrity of the tissue and preserve the biological material for further analysis, all collected kidney samples were stored at a temperature of −80 °C.

## Figures and Tables

**Figure 1 ijms-24-14051-f001:**
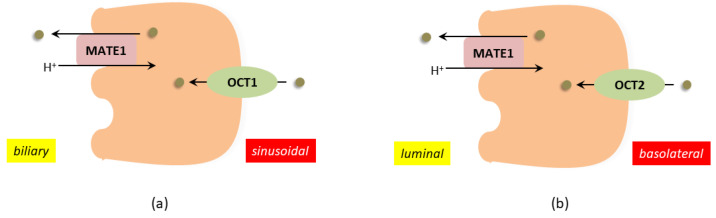
(**a**) Illustrates the organic cation transport systems in human hepatocytes, while (**b**) shows the same in human renal proximal tubules. In both panels, green dots represent organic cations. In the liver (panel (**a**)), hepatocytes express OCT1, located in the sinusoidal membrane domain, facilitating the uptake of organic cations depending on their electrochemical gradient. On the other hand, MATE1 plays a major role in secreting organic cations into bile, functioning as an H^+^/organic cation exchanger. Moving to the kidney (panel (**b**)), proximal tubular cells contain OCT2 in the basolateral membrane domain, facilitating the uptake from the blood of organic cations based on their electrochemical gradient. For the excretion of organic cations into urine, luminally MATE1 primarily serves as an H^+^/organic cation exchanger. Please note that other transport systems are excluded from this figure for simplicity. Modified from [6].

**Figure 2 ijms-24-14051-f002:**
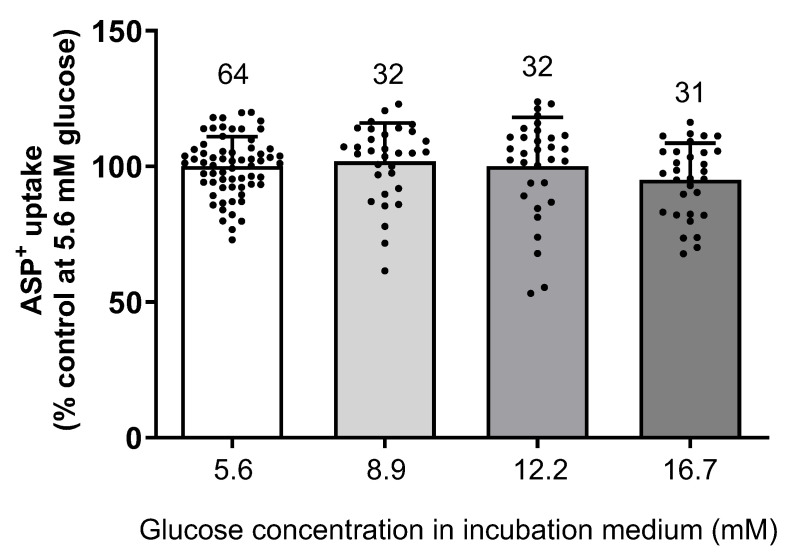
Effect of 24 h incubation with increasing glucose concentrations (5.6–16.7 mM) on the initial uptake of 1 µM ASP^+^ in hOCT2-HEK293 cells. Before measuring ASP^+^ uptake, the incubation solution was replaced by a Ringer-like solution (RLS). The numbers above the columns indicate the number of replicates measured in at least 3 independent experiments. The results of individual measurements are also indicated by a dot. No effect of increasing glucose concentration on ASP^+^ uptake by hOCT2 was observed.

**Figure 3 ijms-24-14051-f003:**
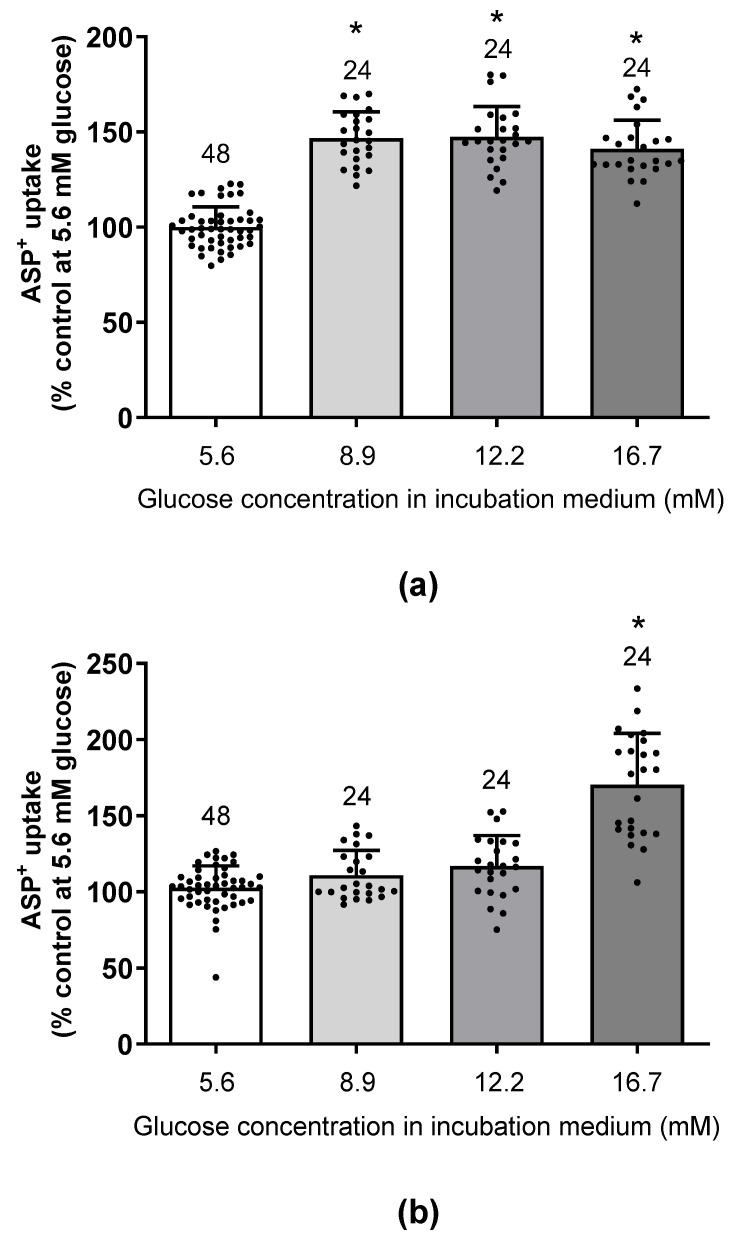
Effect of 48 h incubation with increasing glucose concentrations (5.6–16.7 mM) on the initial uptake of 1 µM ASP^+^ in hOCT2-HEK293 cells (**a**) and of 10 µM ASP^+^ in hMATE1-HEK293 cells (**b**). Before measuring ASP^+^ uptake, the incubation solution was replaced by RLS. The numbers above the columns indicate the number of replicates measured in at least 3 independent experiments. The results of individual measurements are also indicated by a dot. Compared to 5.6 mM glucose, all the other glucose concentrations used significantly stimulated ASP^+^ uptake by hOCT2 (* = *p* < 0.05, ANOVA test with Tukey’s multiple comparison). Compared with all the other glucose concentrations, ASP^+^ uptake by hMATE1 was only significantly stimulated by 48 h incubation with 16.7 mM glucose (* = *p* < 0.05, ANOVA test with Tukey’s multiple comparison).

**Figure 4 ijms-24-14051-f004:**
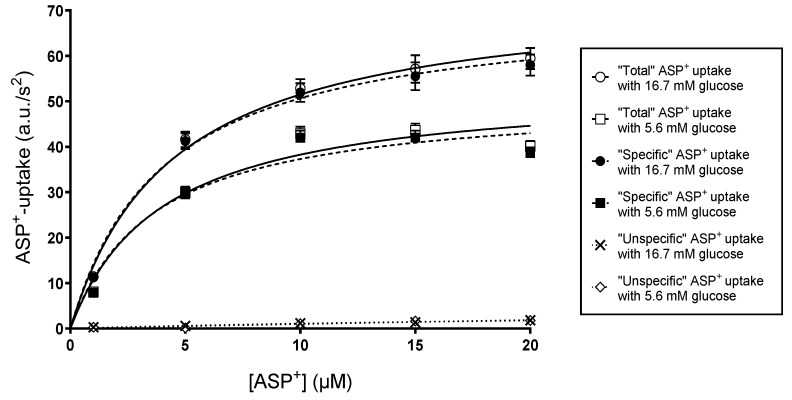
Determination of ASP^+^ specific uptake rates of in hOCT2-HEK293 cells after 48 h incubation with 5.6 (closed squares) or 16.7 (closed circles) mM glucose. Specific ASP^+^ initial uptake rates (dashed curves) were calculated by subtracting the “non-specific” uptake (dotted lines) evaluated as uptake in the presence of 1 mM TPA^+^ as hOCT2 inhibitor (“non-specific” uptake, open diamonds and crosses for incubation with 5.6 or 16.7 mM glucose, respectively) from the total ASP^+^ uptake at 37 °C (continuous solid lines, open squares and open circles for incubation with 5.6 or 16.7 mM glucose, respectively). Values are mean ± SEM of initial fluorescence increase in arbitrary units (a.u.)/s^2^.

**Figure 5 ijms-24-14051-f005:**
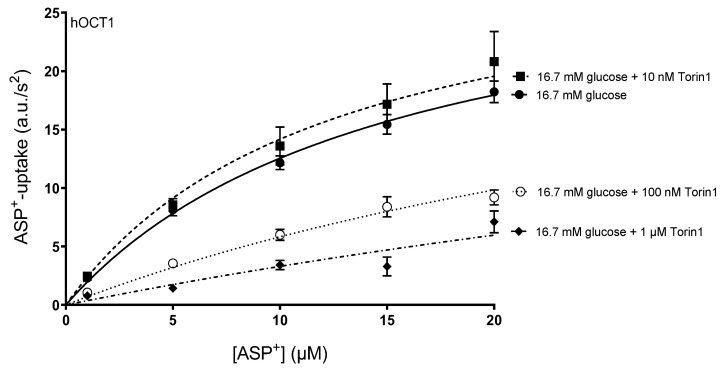
Determination of total ASP^+^ uptake rates in hOCT1-HEK293 cells after 48 h incubation with 16.7 mM glucose (closed circles) in the presence of 10 nM (closed squares), 100 nM (open circles) and 1 µM (closed diamonds) Torin-1. Values are mean ± SEM of initial fluorescence increase in arbitrary units (a.u.)/s^2^.

**Figure 6 ijms-24-14051-f006:**
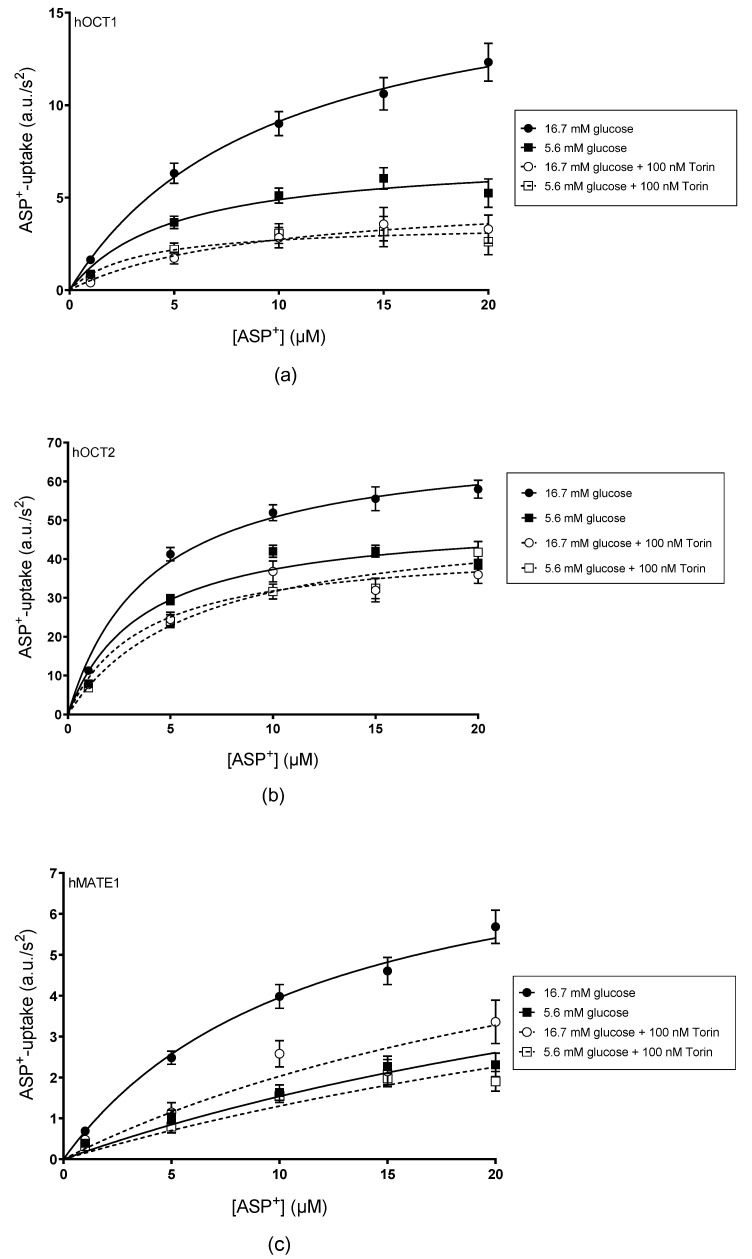
ASP^+^ specific uptake rates in hOCT1- (**a**), hOCT2- (**b**) and hMATE1- (**c**) HEK293 cells after 48 h incubation with 5.6 (closed squares) or 16.7 mM glucose (closed circles) compared to those measured in the presence of 100 nM Torin-1 (open squares for measurements under 5.6 mM glucose and open circles for measurements under 16.7 mM glucose). Values are mean ± SEM of initial fluorescence increase in arbitrary units (a.u.)/s^2^ calculated from 19–63 replicates/concentration measured in at least 3 independent experiments.

**Figure 7 ijms-24-14051-f007:**
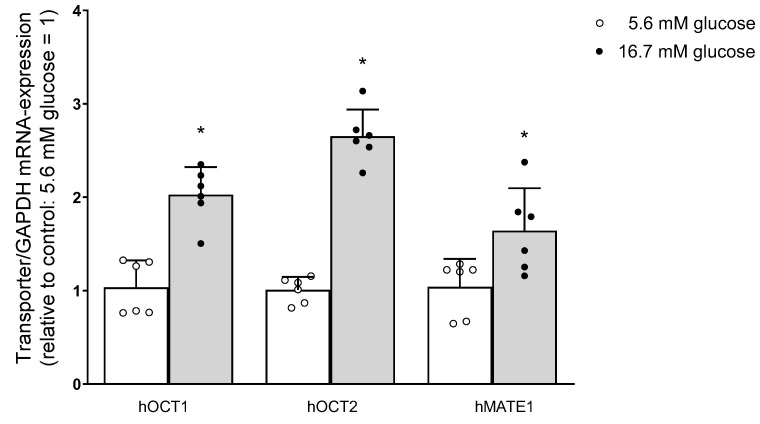
The mRNA levels of hOCT1, hOCT2, and hMATE1 are expressed relative to mRNA GAPDH expression as 2^−ΔCt^, as determined by real-time PCR analysis. Transporter/GAPDH mRNA expression after 48 h incubation with 5.6 mM glucose is set to 1 (open columns). The results of individual measurements are also indicated by a dot. Transporter mRNA expression after 48 h incubation with 16.7 mM glucose (grey columns) was significantly increased (* = *p* < 0.05, Student’s *t*-test) compared to that measured under 5.6 mM glucose as measured in 6 replicates in 3 independent experiments.

**Figure 8 ijms-24-14051-f008:**
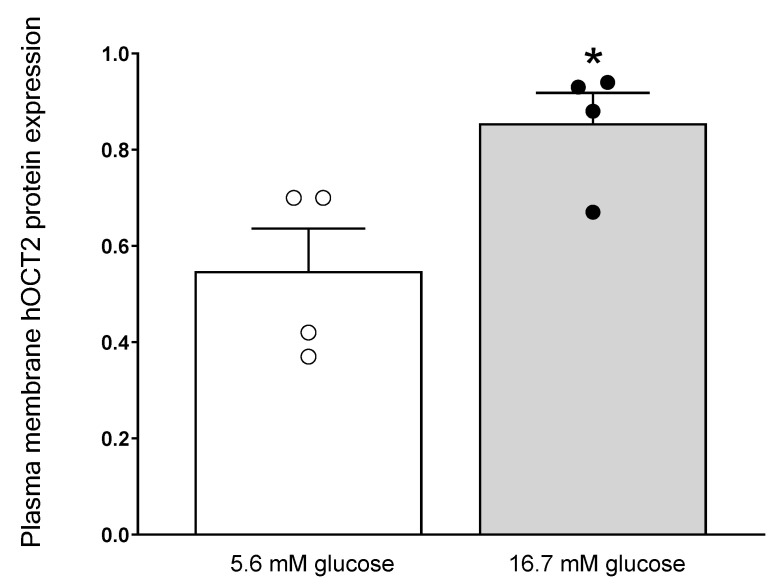
Plasma membrane localization of hOCT2 after 48 h incubation with 5.6 or 16.7 mM glucose. Relative quantification of hOCT2 expression in biotinylated and total fractions from hOCT2-HEK293 cells exposed to 5.6 (open column) or 16.7 mM (grey column) glucose for 48 h. The results of individual measurements are also indicated by a dot. The hOCT2 expression in the plasma membrane after 48 h incubation with 16.7 mM glucose was significantly increased (* *p* < 0.05, unpaired Student’s *t*-test) compared to that measured under 5.6 mM glucose as measured in 4 independent experiments.

**Figure 9 ijms-24-14051-f009:**
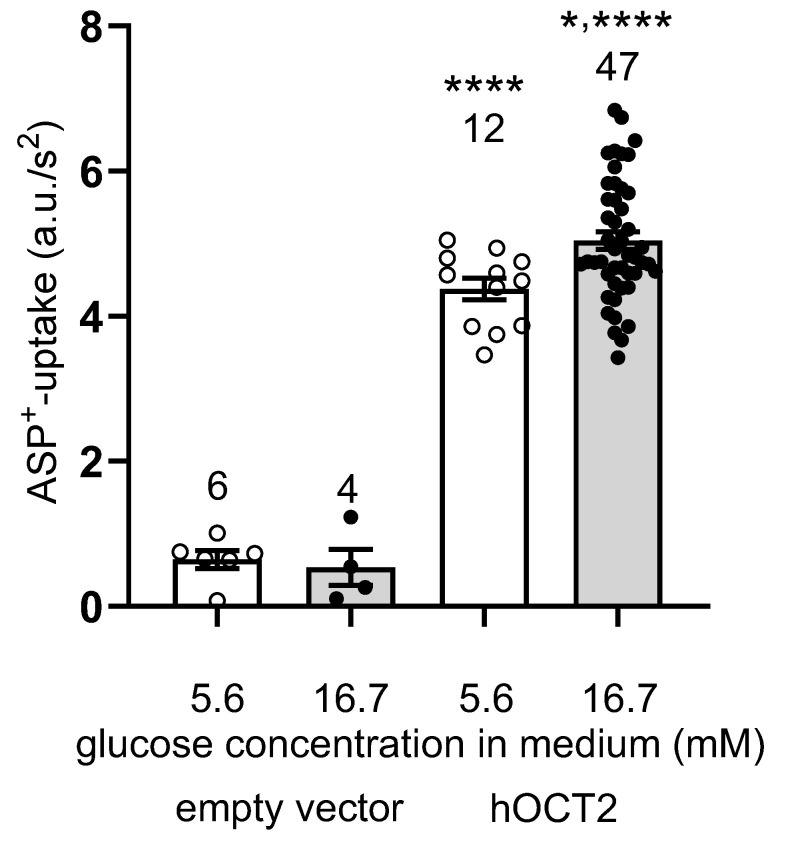
Effect of 48 h incubation with 5.6–16.7 mM glucose concentrations on initial uptake of 4 µM ASP^+^ in MDCK cells expressing the empty vector or hOCT2. Before measuring ASP^+^ uptake, the incubation solution was replaced by RLS. The numbers above the columns indicate the number of replicates measured in at least 3 independent experiments. The results of individual measurements are also indicated by a dot (open dots for 5.6 mM glucose, closed dots for 16.7 mM glucose). Compared to MDCK cells expressing the empty vector, hOCT2-expressing cells have a significantly higher ASP^+^ uptake (**** *p* < 0.0001, ANOVA test with Tukey’s multiple comparison test). Compared with 5.6 mM glucose, 16.7 mM glucose significantly stimulated ASP^+^ uptake by hOCT2 (* *p* < 0.05, ANOVA test with Tukey’s multiple comparison test).

**Figure 10 ijms-24-14051-f010:**
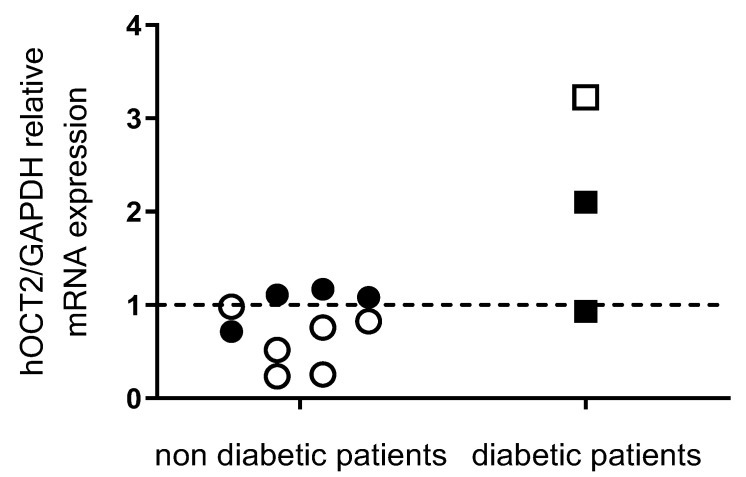
The mRNA content of hOCT2 in human kidneys is expressed relative to the mRNA expression of GAPDH as 2^−ΔCt^, as determined by real-time PCR analysis in samples from male (closed symbols) and female (open symbols) non-diabetic (circles) and diabetic (squares) patients. The mean hOCT2/GAPDH mRNA expression in male patients was set to 1 (interrupted line). No significant difference in hOCT2-mRNA expression was observed between the kidneys of male and female non-diabetic patients. The results obtained with samples from diabetic patients suggest a trend towards higher expression of hOCT2 in the kidneys compared to what was observed in non-diabetic patients. No statistical analysis was performed due to the small number of patients in this group. Each point represents a sample from one patient.

**Table 1 ijms-24-14051-t001:** K_m_ (µM) and V_max_ (a.u./s^2^) calculated from the ASP^+^ uptake saturation experiments with hOCT1-, hOCT2-, and hMATE1-HEK293 cells after 48 h incubation with 5.6 or 16.7 mM glucose and in the presence or absence of 100 nM Torin-1. Values are expressed as mean ± SEM.

	hOCT1		hOCT2		hMATE1	
	K_m_ (µM)	V_max_ (a.u./s^2^)	K_m_ (µM)	V_max_ (a.u./s^2^)	K_m_ (µM)	V_max_ (a.u./s^2^)
5.6 mM glucose	5 ± 2	7 ± 1	4 ± 2	51 ± 6	13 ± 5	4 ± 1
16.7 mM glucose	10 ± 1	18 ± 1 *	4 ± 1	71 ± 4 *	14 ± 3	10 ± 1 *
5.6 mM glucose + 100 nM Torin-1	3 ± 2	4 ± 1	6 ± 2	51 ± 6	12 ± 6	3 ± 1
16.7 mM glucose + 100 nM Torin-1	9 ± 4	5 ± 1	4 ± 2	43 ± 5 #	16 ± 16	6 ± 3

An asterisk indicates a statistically significant difference from all the other values obtained for the specific transporter type (* = *p* < 0.05, ANOVA test with Tukey’s multiple comparison); # indicates a statistically significant difference from experiments after 48 h incubation with 16.7 mM glucose alone (# = *p* < 0.05, unpaired Student’s *t*-test).

## Data Availability

No new data were created or analyzed in this study. Data sharing is not applicable to this article.

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
