# Peer review of "Regulation of Transporters for Organic Cations by High Glucose"

_ijms, 2023, doi:10.3390/ijms241814051_

Round 1

Reviewer 1 Report

Comments:

-       In figure 2, the authors should adjust the title of the x-axis. Same comment for figure 3a and 3b.

-       Can the authors generate two additional figures:

o   The uptake of ASP+ as a function of the concentration of ASPafter 48h incubation with 5.6 mM glucose and in the presence of hOCT1, hOCT2, hMATE, and in the presence and absence of Torin-1

o   Same concept but with the incubation of 16.7 mM

These figures are a summary of Table 1, and it allows for the reader to have a better understanding of the kinetics.

-       “Incubation with 16.7 mM glucose significantly increased the Vmax of hOCT1, hOCT2, and hMATE1, without changing their affinity for transporting the fluorescent organic cation ASP+”: As much as I agree with the statement, yet for hOCT1, the affinity was divided by 2 (since Km doubled) when the concentration of glucose increased from 5.6 to 16.7 mM. Therefore, I advise the authors to take this point into consideration and rephrase the sentence.

Questions:

-       Following the experiment assessing the specific uptake of ASP+, it shows that there is very little unspecific uptake of ASP+ in the hOCT2-HEK293 cells. Based on my knowledge, rarely can we see this kind of phenomenon in biological experiments in which the majority of the uptake is specific (approximately 98%). Can the authors explain why ASP+ uptake is so specific?

-       Why when you incubated the cells with 1µM of ASP+, there was a significant increase of the uptake of ASPat 8.9 mM of glucose, whereas when the cells were incubated with 10µM of ASP+, the increase only appeared at 16.7 µM? Is there any mechanisms that can explain this observation?

-       In figure 5, why there is more uptake of ATP+ in the presence of hOCT2 compared to hOCT1?

-       In figure 5, why when adding 10nM of Torin-1, the rate of the specific binding of ATP+ increases compared to the control curve (without Torin-1)?

-       Why the authors did not measure the expression of hOCT1, hOCT2 and hMATE mRNA levels in the presence of Torin-1 to confirm what they have seen in the uptake experiments?

-       In Ma, L. et al. (2009), it was found that OCT1 was phosphorylated and thus the affinity increased to the substrate. You have mentioned this finding in your discussion. This said, why do you think that your findings is not the same of the ones observed by Ma, L. et al.?

-       Does these transporters can be upregulated or downregulated by the increase of Reactive oxygen species induced by hyperglycemia? 

-       Does these transporters expression are in relation with other transporters like SGLT1, SGLT2, GLUT 2 or other transporters?

-       “Consequently, there is a risk of metformin accumulating in the liver due to the possible glucose-induced up-regulation of OCT1”. I do not agree with the statement. Metformin is a drug that is know that it improves glucose transport in hepatocyte and there is an extensive literature about this matter. Therefore, I can’t understand how the up-regulation of OCT will affect the action of metformin since there are other transporters that can reduce the concentration of glucose and thus not inducing an accumulation of metformin? Can the authors discuss this matter.

-       The authors should revise the abstract, the introduction and the discussion section. I suggest that they should shorten the long phrases so it could be clearer to the reader.

Reviewer 2 Report

In this manuscript, the authors investigated the effect of high-concentration glucose on the expression level of Organic Cations transporters. They determined the transport activity of the transporter with a cell-based assay and the mRNA level of the transporters with the presence of different concentrations of glucose. The results indicate that incubation of a high concentration of glucose can increase the expression level of transporters, therefore leading to a higher transport signal. 

To make the manuscript more valid, I have the following questions about the paper:

1. The ASP transport level was determined in a cell-based assay, I wonder if the author considered the difference in cell growth rate in different concentrations of glucose. The level of transport should be calibrated with the number of cells in each group. The difference in the 24h and 48h groups could be caused by the difference in cell density rather than the expression level difference. 

2. To validate the effect of high glucose on transporter expression, the author quantified mRNA levels in different conditions. I believe western blot will be a better method to show the protein level difference with/without glucose and treatment of 24/48 hrs. Therefore, I suggest the author use western blot with better controls (such as actin) to show the difference in protein expression level. A control is also needed for the biotinylation result. 

3. For all the bar grams, it will be helpful to show the single data spots on it. It will make readers better understand the distribution of the repeats. 

4. Also, please include all raw data about Km/Vmax of ASP transport in the supplementary file. 

Round 2

Reviewer 2 Report

The author modified the manuscript according to my suggestions and most of the questions raised were answered. Though I still believe a calibration or a control for the biotylation experiment is needed to make sure the overall protein level used in each group is similar, but current results can be used to support their conclusion. 

Author Response

We thank this reviewer for the suggestion. Unfortunately, we did not perform a calibation curve for the biotinylation experiments and have used the current results to support the conclusions